# Pet-Human Gut Microbiome Host Classifier Using Data from Different Studies

**DOI:** 10.3390/microorganisms8101591

**Published:** 2020-10-15

**Authors:** Nadia Bykova, Nikita Litovka, Anna Popenko, Sergey Musienko

**Affiliations:** Atlas LLC, Malaya Nikitskaya 31, 121069 Moscow, Russia; litovka@atlas.ru (N.L.); popenko@atlasbiomed.com (A.P.); musienko@atlasbiomed.com (S.M.)

**Keywords:** gut microbiome, host classification, random forest

## Abstract

(1) Background: microbiome host classification can be used to identify sources of contamination in environmental data. However, there is no ready-to-use host classifier. Here, we aimed to build a model that would be able to discriminate between pet and human microbiomes samples. The challenge of the study was to build a classifier using data solely from publicly available studies that normally contain sequencing data for only one type of host. (2) Results: we have developed a random forest model that distinguishes human microbiota from domestic pet microbiota (cats and dogs) with 97% accuracy. In order to prevent overfitting, samples from several (at least four) different projects were necessary. Feature importance analysis revealed that the model relied on several taxa known to be key components in domestic cat and dog microbiomes (such as Fusobacteriaceae and Peptostreptococcaeae), as well as on some taxa exclusively found in humans (as Akkermansiaceae). (3) Conclusion: we have shown that it is possible to make a reliable pet/human gut microbiome classifier on the basis of the data collected from different studies.

## 1. Introduction

A microbiome is a complex ecological structure that is unique to each environment. Microbiota inhabiting living organism sites, such as the human gut, are of particular interest. Even though metagenomic approaches have made revealing microbiome compositions routine, their characterization and the identification of unique traits is still a challenge.

In the field of microbiome classification, there are several classification settings. One is the classification of the microbiome as a whole. There are also models for disease prediction, and some other individual trait predictions, such as age [1] or specific owner identification (skin microbiome), [2] for example. The gut microbiota, despite its complexity and great variation between individuals, was shown to be predictive of various intestinal diseases and conditions, such as irritable bowel syndrome (IBS) [3], Crohn’s disease [4,5], and colorectal cancer [6]. Interestingly, the composition of the gut microbiome also predicts some non-intestinal illnesses, such as coronary artery disease [7], liver fibrosis [8], metabolic diseases/obesity [9], insomnia [10], and bipolar depression [11]. Another classification setting is the detection of contamination in samples. This task mostly arises in the case of water contamination with sewage or animal faeces. In this setting, the fraction of sequences coming from another host can be minimal. Moreover, contamination might come from several different sources making this task quite challenging [12,13,14]. In our study, we had a specific goal: host discrimination of the whole gut microbiome. This task arose from our need to filter out faecal samples of pets mistakenly or intentionally sent for commercial microbiome analysis in guise of human ones.

The mammalian gut microbiome evolves together with the host. As a general rule, the distance between microbiomes increases with the evolutionary distance between hosts [15]. At the same time, microbiome composition is also a reflection of the host’s dietary category. For example, while sharing the same main bacterial phyla, herbivores and carnivores harbor different families of Clostridiales (Ruminococcaceae and Peptostreptococcaceae, respectively) [15]. Herbivores also show greater diversity at all taxonomic levels than carnivores. It is also generally accepted that species with the same diet are similar at a higher taxonomic level, while host phylogeny reveals itself more at a lower taxonomic level (species and strains) [15,16]. Accordingly, this information was used in host prediction (contamination prediction) models. The oligotyping technique harnesses the existence of the host-specific/preferred species and strains, showing that it is possible to detect sequences from a specific host using just one or several genera [12,13]. This analysis is also considered to be more robust to high fluctuations in microbiome taxonomic composition caused by different sequencing techniques. The total microbiome taxonomy groups can be used as well, for example, as an input to the state-of-art program for contamination tracking called Source Tracker [14]. A disadvantage of Source Tracker for host classification is that it requires host data training from the same experiment to build a prediction model. Here, we checked if it is possible to use regular taxonomy features for accurate microbiome classification of pets and humans.

It is a widely acknowledged problem that next-generation sequencing (NGS) data in general, and microbiome 16S rRNA sequencing data specifically, vary from data center to data center. This introduces strong batch effects that sometimes make meta-analysis a rather sophisticated statistical task [17,18]. This is caused by different protocols for sample collection and storage [19], different DNA purification [20] and amplification protocols [21], and the use of different sequencing platforms [22]. Moreover, sequencing of different regions of the 16S rRNA gene influences the abundance of specific taxa in the resulting data [23]. A recent meta-analysis of human microbiomes found that differences in experimental protocols can affect microbiome composition more than the biological variance of some traits [17]. Another meta-analysis, this time of colorectal cancer studies, demonstrated that samples clustered primarily by study [18]. On the other hand, the incorporation of several studies helped increase the overall accuracy in this study. It should be noted that, in this meta-analysis, each study still had its own set of control (healthy) samples. Here, we aimed to build a classifier using public data where each class (host) was sequenced in a separate study, as there were no studies where human and pet samples were sequenced in the same experiment.

Random forest (RF) algorithm [24] was chosen for the classification task. RFs are highly used machine learning algorithms for microbiome classification [3,4,5,6,7,8,9,10,11] due to the limited number of model parameters and simple results interpretation.

## 2. Materials and Methods

### 2.1. Data

Data with publicly available cat, dog, or human faecal samples used in our study are listed in Table 1. Only projects that performed Illumina sequencing of the V4 region of 16S RNA were included. Overall, data from five cat projects and seven dog projects were collected, providing 321 pet samples in total. Ten human projects provided 1242 samples. Note that from the specified human projects, we used only healthy control (HC) subjects, and from the pet projects we used all subjects (i.e., not only healthy samples) to provide a bigger dataset. After the models were trained, we further tested them on five independent pet and two independent human projects. These additional projects contained 432 animal and 358 human samples.

### 2.2. QIIME2

All the raw data fastq files were processed with QIIME 2 (Flagstaff, AZ, USA) [47] to obtain Chao diversity estimations [48] and feature tables. The following parameters were employed in the QIIME2 microbiome analysis pipeline:DADA2 [49] denoising quality parameter value was set to 10 (--p-trunc-q 10),taxonomy assignment using QIIME2 feature-classifier [50],a random subsample of 5000 reads was used to calculate feature tables,alpha-diversity was calculated by sampling 5000 random reads five times from the whole sample to decrease the impact of low-abundance bacteria; the resulting chao index is the mean of these iterations results,a custom reference database was used; the database is a restricted version of SILVA database [51], aligned to the HITdb database [52] in order to leave mostly gut bacteria.

### 2.3. Grouping Features at Different Taxonomic Levels

Abundance data at a genus level was used in the analysis. Each genus is represented as its full taxonomy, namely: kingdom, phylum, class, order, family, genus. Therefore, the original genus table can be grouped into higher-level features. During this procedure, the abundances of genera grouped by the same higher taxa were summed. The analysis was then performed for the tables at each taxonomic level.

### 2.4. Filtering Rare Features

Statistical hypothesis testing and RF training were only performed for the taxa left after filtering out features with low abundance. The taxonomic features where over 90% of samples had zero read for both pet and human data were defined as rare. This procedure reduced the initial 386 genera to 138, 123 families to 55, 54 orders to 29, 26 classes to 18, 15 phyla to 10 (Appendix A).

### 2.5. Mann-Whitney Test

Two-sided Mann-Whitney test was performed on the projects’ median abundance values for each feature (i.e., for each feature there were 12 pet values versus 10 human values in the test). We applied both Holm–Bonferroni [53] and FDR Benjamini–Hochberg [54] procedures to correct for the testing of multiple features. Significant features were further used to build restricted versions of the RF models. Two-sided Mann-Whitney test was also applied for the Chao diversity values that characterize each sample.

### 2.6. t-SNE

t-SNE algorithm from Python3 sklearn library [55] with Bray-Curtis distance was used to visualize the data. The analysis was performed for the balanced class dataset (see below). The input taxa were restricted to 138 most abundant genus (see above).

### 2.7. Balanced Class Dataset

From the initial dataset that contained different numbers of animal and human samples, and different numbers of samples from different projects, we constructed a dataset balanced by host (321 animal and 321 human samples). It included all our animal data and a subset of human data. The accession numbers of specific samples that fell into the dataset are listed in Appendix A. The human subset was balanced by projects (i.e., we aimed to take the same number of samples from each project). The sampling was without replacement. This dataset was also used for data visualization using t-SNE (see above).

### 2.8. CLR (Center Log Ratio) Transformation

CLR transformation of feature tables was performed with Python3 library scikit-bio version 0.5.6 (http://scikit-bio.org). CLR transformation converts compositional data from Aitchison geometry to the real space [56]. Non-transformed or CLR-transformed data were optionally used to train the RF model.

### 2.9. RF Implementation

The input data matrix for the model consisted of feature abundance values and a column with mean Chao diversity values. Optionally, the data were restricted to fewer features, or CLR-transformed data was used beforehand. To construct an optimal prediction model, we first performed parameter selection using stratified 5-fold cross-validation. The best parameters were defined by the highest average test accuracy achieved at cross-validation. We varied the following model parameters:max_features in the range from 2 to the ‘number of features’,max_depth in the range from 2 to 52,min_samples_split in the range from 2 to 52,n_estimators from the set {1,5,10,50,100,500,1000}.

To speed up the parameter selection process, we performed it in two steps. First, we selected the best set of parameters using all parameter combinations in a smaller range. During the second step, each parameter was refined while all the other parameters were fixed to the values obtained on the first step (Text S1). The model was then fit on the whole dataset with the best parameters, and out-of-bag scores were reported as a final performance estimation of the model. The project is realized using Python 3 sk-learn library. The source code of the project is available at GitHub (https://github.com/nadiabykova/microbiota_host_classifier).

### 2.10. The Project Learning Curve

Our models are trained on data from specific projects. Therefore, it is possible that the model ‘remembers’ project features rather than host-specific features. If so, new projects that did not participate in the training would be poorly predicted. To evaluate the ability of these models to overfit to specific projects, we conducted the following experiment: we varied the number of human projects incorporated in the training set and measured the performance on the projects that did not participate in the training. Formally, for *n* from 1 to *N* − 1, where *N* is the number of human projects, we formed new balanced training sets consisting of all animal data and samples from n human projects. The number of considered project combinations for each *n* was set to the minimum value of 200 and C_N−1_^n^. The training sets were balanced by class, and the human part of the sets was balanced by project. Here, we used sampling with replacements to be able to construct the human part of the set with size 321 for each *n*. We also randomized the sampling from the human projects taking five random samples for each combination of projects. Therefore, in total, approximately 200 × 5 models were trained for each value of *n* (less for the cases where *n* < 4). Accuracy on projects that did not participate in the training was then evaluated.

## 3. Results

### 3.1. t-SNE Plot

To visualize our data, we first built a t-SNE plot on the balanced class dataset (i.e., a sample of original data that contains equal amounts of pet and human samples (see Materials and Methods)). Figure 1 shows that human, cat, and dog samples cluster within groups, indicating that they can possibly be classified using taxonomic features’ abundance values. Samples from specific studies also tend to cluster together, but the host signal is stronger.

### 3.2. Taxa Differentially Abundant in Pets and Humans

We studied the distribution of taxa abundances in pets and humans at several taxonomic levels (see Materials and Methods). First of all, the data show substantial variation inside host groups. Even on the phylum level, ‘project-outliers’ can be noted, illustrating that the batch effect can significantly affect the abundance of specific taxa (Appendix A). To detect the taxa differently abundant in pets and humans, we applied two-sided Mann-Whitney test to the projects’ median values of each taxon abundance at each level. The significant taxa detected are listed in Appendix A. On the phylum level, Verrucomicrobia and Fusobacteria were significant (Appendix A), which is in accordance with previous studies describing Fusobacteria as one of the key phyla of domestic pets, and Verrucomicrobia as a taxon characteristic of the human microbiota while being absent in cats and dogs [57]. On the level of class, Verrucomicrobia, Fusobacteria, and Deltaproteobacteria were detected; Deltaproteobacteria was present in almost all human projects, and absent or present in very small amounts in pet projects (Appendix A). On the level of order, Verrucomicrobiales and Desulfovibrionales were detected (Fusobacteriales were detected only by the FDR correction method, Appendix A). On the family level, there were more interesting results (Figure 2). While the same effect at the higher levels was detected for Akkermansiaceae and Desulfovibrionaceae, the difference in Bacteroidetes and Firmicutes was first detected at the family level. Namely, the Peptostreptococcaceae family was characteristic of pets and Ruminococcaceae family for humans; this switch in the usage of Clostridia families was previously described as a difference between carnivorous and herbivorous animals [15]. Peptostreptococcaceae is also described as a prominent taxon for cats [43]. The families of the Bacteroidetes phylum, Rikenellaceae and Marinifilaceae, were detected almost exclusively in humans. FDR correction added another eight families from different phyla. On the genus level, Astilipes and Odoribacter from the Rikenellaceae and Marinifilaceae families, respectively were detected, Ruminococcus_1 and Faecalibacterium (elevated amounts in humans) from Ruminococcaceae family, Bilophila from Desulfovibrionaceae family, Akkermansia, 5 Lachnospiraceae genera, and Erysipelotrichaceae_UCG-003 were detected (Appendix A). FDR correction yielded an additional 20 significant genera from different taxa. Taken together these data show that, at the level of general composition, pets and humans do not show a great difference with all the main phylums being insignificant (note that the Verrumicrobia and Fusobacteria consist of only one or two genera), while all the main differences lay at the level of families and specific genera. The defined significant taxa were further used to build restricted RF models. Mann-Whitney test for the median values of the Chao diversity index was insignificant.

### 3.3. Random Forest Models

RF models were trained on the dataset balanced by the host class that was derived from the initial dataset (see Materials and Methods). To define the most appropriate model for host discrimination, we tested several types of models. We tried models on different taxonomic levels, with all or only specific sets of features (as defined by Mann-Whitney test above), we also optionally applied CLR transformation to the initial data (see Materials and Methods). For each model, the best model parameters were first defined using cross-validation. The models with the best parameters were fit on the dataset, and performances were estimated on out-of-bag (OOB) samples. The obtained best parameters, cross-validation and out-of-bag results are summarized in Appendix A and Table 2. Very good results were achieved by all the models. However, genus models clearly outperformed family models, and genus models with more features (Genus_ALL and Genus_MW-FDR) were better than the most restricted Genus_MW-Holm model because a model using more meaningful features allows for better prediction. Remarkably, using only MW-selected FDR features did not lead to a substantial decrease in accuracy. The usage of CLR transformation did not introduce any significant improvement. The fact that the usage of CLR transformation did not significantly improve results in our case might be because the prediction accuracy is good with both methods. The OOB estimation of accuracy of both Genus_ALL and Genus_MW-FDR models was 0.99 ± 0.002. The process of parameter selection for each model and corresponding ROC curve obtained in cross-validation for each model are presented in Text S1. The ROC curves of all the models together are shown in Appendix A.

### 3.4. Random Forest Feature Importance

RF models are suitable for selecting the most important features that allow us to distinguish between classes. When comparing the most important RF features of full models (Family_ALL and Genus_ALL) with MW-results (FDR correction), we found good correspondence both on the family level (86% intersection between top features, Appendix A) and genus level (78%, Appendix A). New families preferred by RF over Mann-Whitney-selected taxa were Enterococcaceae and unclassified bacteria. At the genus level, the RF model brought up Fusobacterium, Collinsella, Anaerobiospirillum, unclassified Fusobacteriaceae, unclassified bacteria, Intestinimonas, Veillonella.

### 3.5. Projects Learning Curve

We further tried to estimate if models built this way are robust for project overfitting. To this end, we conducted the following experiment: the pet part of the dataset was fixed, and from the human data we selected various numbers of projects to include in the training set. The remaining human projects were used to control the model’s performance (see Materials and Methods). The dependency of model accuracy on the number of human projects in the training set for genus models is shown in Figure 3 (the accuracy is averaged among the combinations of training projects and among the projects used for testing). Figures for the models where accuracy is shown for each test project can be found in Appendix A. Figure 3 shows that models trained using only one human project appear to consistently overfit, leading to decreased accuracy on the test projects. The appropriate number of projects to capture the host differences for our setting should be more than four (as can be identified from the graph). In the models described above, we used more projects (all of them) in the training set, thus we do not expect them to be overfit. For the Genus_MW-FDR model, we do not expect accuracy below 0.9 on other human projects (Appendix A).

### 3.6. Model Testing on Additional Projects

To test the performance of our actual models on some new projects, we downloaded several additional human and animal projects (described in Table 1) and applied our models to them. The resulting accuracy for all genus models is summarized in Appendix A. The best results were once again shown by Genus_ALL and Genus_MW-FDR models. Genus_MW-FDR model performance is presented in Table 3 and Figure 4. The new data also contained samples from SLE (systemic lupus erythematosus) and RA patients (rheumatoid arthritis); notably, all the models showed better performance on healthy samples, as expected. From animal projects, almost all new samples were dog samples –only two projects also contained cat samples. The dog samples were better recognized as pets by all the models. To obtain the full discriminative characteristics of the models, we constructed 100 balanced animal/human sets of size 200 from these projects (50 cats, 50 dogs, and 100 humans). The average accuracy, precision, recall, and f1 score of the Genus_MW-FDR model on healthy samples is 0.99 ± 0.01, 0.98 ± 0.02, 1.00 ± 0.0, 0.99 ± 0.01 on all samples 0.97 ± 0.01, 0.98 ± 0.02, 0.97 ± 0.02, 0.97 ± 0.01 (Table 4), the estimations for the other genus models are in Appendix A.

## 4. Discussion

One result of our work was identifying the list of taxa differently abundant in pets and humans that is statistically supported by a set of different studies. We show that the main differences do not occur in the general composition of the microbiome, but rather in the usage of specific genera, even though the species stick to very different diets. The main, if not to say the only, Verrumicrobia member in a gut microbiome is Akkermansia genus. Akkermansia bacteria are known to live in a mucin layer and degrade it. Previous studies show that indeed, feline and canine gut microbiome lack Akkermansia because this ecological niche is occupied with other species, specifically members of Bacteroidaceae, Prevotellaceae, Clostridiales, Faecalibacterium, and Fusobacteria phylum [58]. We also show that, even when overfitting occurs (as observed in a few projects in the training set), the RF model appears to be able to dissect host-specific features from the project-specific features, and the joint usage of these features makes it possible to successfully classify samples from new projects. On the other hand, even though we collected data from a substantial number of projects, we cannot guarantee that all the possible sequences techniques were covered and that our model will not fail at some special new project due to skewed taxa abundances. Moreover, samples from people with health conditions are more likely to be mistaken as pet samples.

## 5. Conclusions

We have shown that it is possible to make a reliable pet/human classifier on the basis of taxonomic feature abundance tables collected from different studies. Our study demonstrates that a classifier with good performance can be built if at least four different studies are included in the training set. We also provide a list of taxa that discriminate between hosts, these results are in line with previous studies.

## Figures and Tables

**Figure 1 microorganisms-08-01591-f001:**
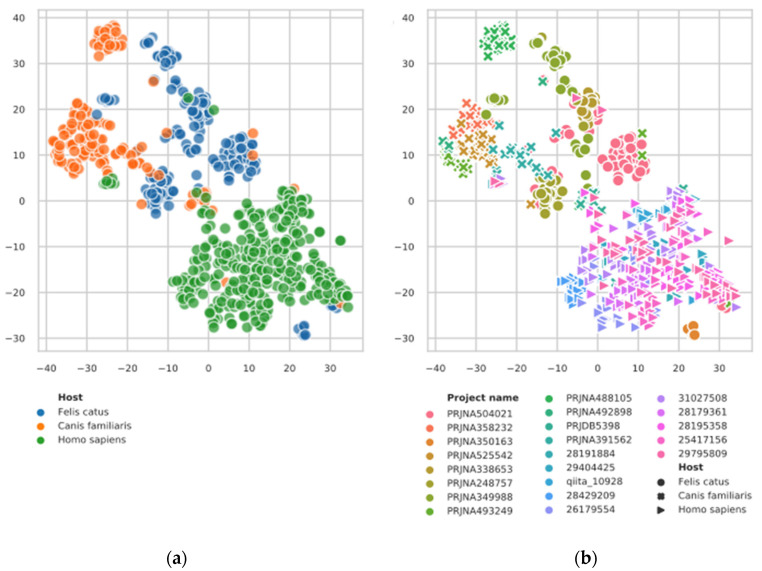
t-SNE plot for the dataset on the genus level. The t-SNE plot was built using 138 most abundant genera of the balanced dataset (see Materials and Methods). The Bray–Curtis dissimilarity between vectors was used. The samples are colored by host (**a**), or by the study name (**b**). On side b, the host is shown by a marker shape. See the in-plot legend for the specific name-color mapping.

**Figure 2 microorganisms-08-01591-f002:**
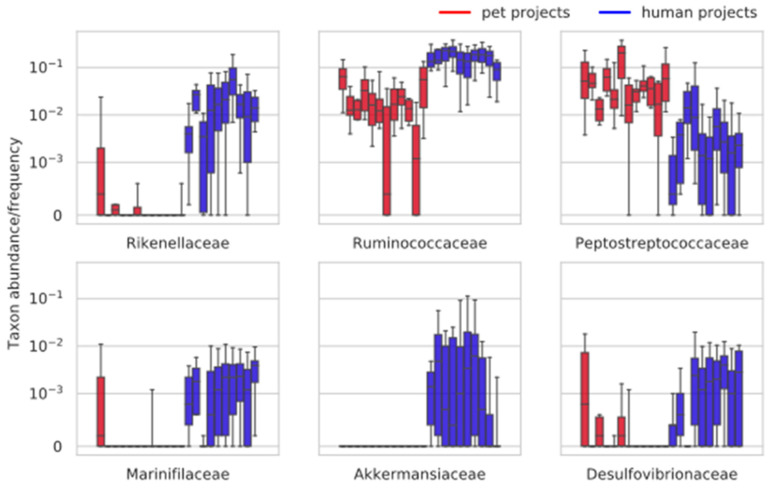
The differentially abundant families between pets and humans. The y-axis shows the fraction of reads from the total (the axis is log-transformed). Red bars correspond to pet projects, and blue bars correspond to human projects. The families significant using the Holm correction method are shown (for families significant according to the FDR correction method, see Appendix A).

**Figure 3 microorganisms-08-01591-f003:**
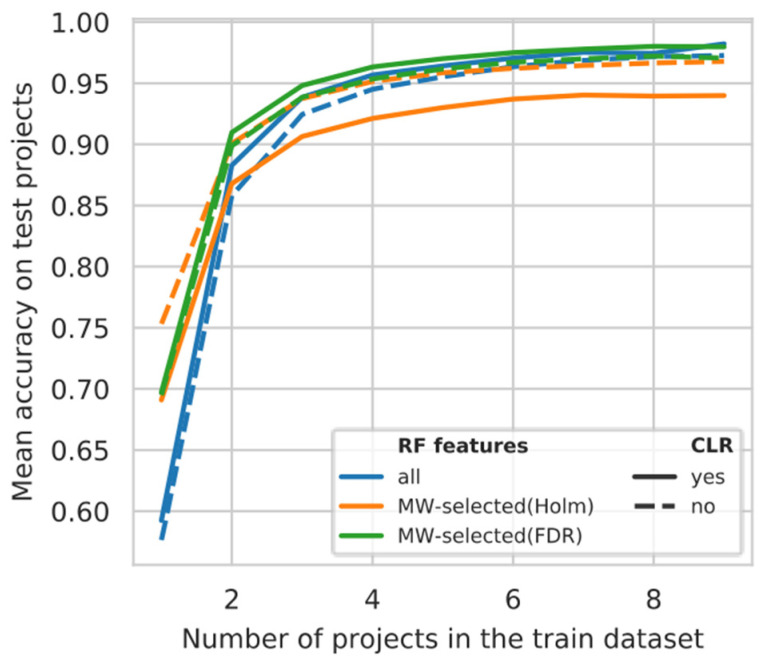
The dependency of accuracy on test human projects from the number of human projects used in the training set (for the genus models). The average value of accuracy among all the models (for the specific *n*) and test projects is shown.

**Figure 4 microorganisms-08-01591-f004:**
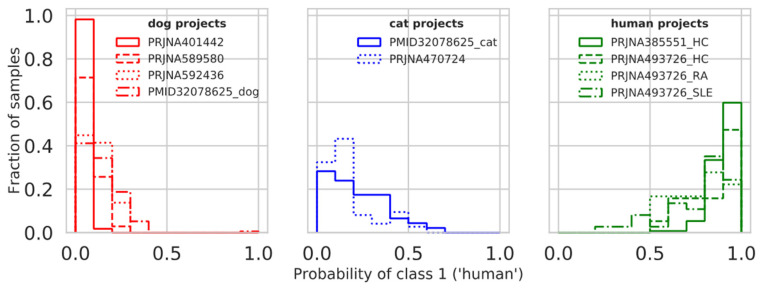
Prediction score distributions of the additional projects. The histograms for the probability of class 1 (‘human’) for the seven additional projects are present. The dog projects are shown in red, cat projects in blue, and human projects are in green. Specific projects are shown with the combination of color and linetype (see the in-graph legend).

**Table 1 microorganisms-08-01591-t001:** Data used in the study.

Project Name	Host	Host Type	Number of Samples	Number of Samples (Train Dataset)	PMID	Author/Year	Ref.
PRJNA504021	Felis catus	pet	65	65	31844119	*Marsilio* et al. *(2019)*	[25]
PRJNA349988	Felis catus	pet	44	44	27912797	*Duarte* et al. *(2016)*	[26]
PRJNA248757	Felis catus	pet	30	30	25279695	*Bell* et al. *(2014)*	[27]
PRJNA338653	Felis catus	pet	19	19	30709324	*Whittemore* et al. *(2019)*	[28]
PRJNA350163	Felis catus	pet	6	6	28278278	*Vientós-Plotts* et al. *(2017)*	[29]
PRJNA488105	Canis familiaris	pet	34	34	no paper		
PRJNA525542	Canis familiaris	pet	32	32	31565574	*Jarett* et al. *(2019)*	[30]
PRJNA358232	Canis familiaris	pet	30	30	no paper		
PRJNA391562	Canis familiaris	pet	23	23	29852000	*Herstad* et al. *(2018)*	[31]
PRJNA493249	Canis familiaris	pet	19	19	32027665	*Fujishiro* et al. *(2020)*	[32]
PRJDB5398	Canis familiaris	pet	13	13	29643280	*Omatsu* et al. *(2018)*	[33]
PRJNA492898	Canis familiaris	pet	6	6	32027665	*Fujishiro* et al. *(2020)*	[32]
PMID29795809	Homo sapiens	human	681	46	29795809	*McDonald* et al. *(2018)*	[34]
PMID25417156	Homo sapiens	human	200	45	25417156	*Goodrich* et al. *(2014)*	[35]
PMID28195358	Homo sapiens	human	115	45	28195358	*Hill-Burns* et al. *(2017)*	[36]
PMID28179361	Homo sapiens	human	102	45	28179361	*Pascal* et al. *(2017)*	[37]
PMID31027508	Homo sapiens	human	49	45	31027508	*Liu* et al. *(2019)*	[38]
PMID26179554	Homo sapiens	human	31	31	26179554	*Keshavarzian* et al. *(2015)*	[39]
PMID28429209	Homo sapiens	human	22	22	28429209	*Petrov* et al. *(2017)*	[40]
qiita_10928	Homo sapiens	human	21	21	no paper		
PMID29404425	Homo sapiens	human	12	12	29404425	*Zhou* et al. *(2018)*	[41]
PMID28191884	Homo sapiens	human	9	9	28191884	*Halfvarson* et al. *(2017)*	[42]
**Additional Projects**							
PRJNA470724	Felis catus	pet	74		29971046	*Bermingham* et al. *(2018)*	[43]
PMID32078625	Felis catus	pet	46		32078625	*Jha* et al. *(2020)*	[44]
PMID32078625	Canis familiaris	pet	192		32078625	*Jha* et al. *(2020)*	[44]
PRJNA401442	Canis familiaris	pet	56		no paper		
PRJNA589580	Canis familiaris	pet	35		no paper		
PRJNA592436	Canis familiaris	pet	29		no paper		
PRJNA385551	Homo sapiens	human	284		28959739	*Bian* et al. *(2017)*	[45]
PRJNA493726	Homo sapiens	human	74		30872359	*Li* et al. *(2019)*	[46]

**Table 2 microorganisms-08-01591-t002:** Out-of-bag estimations of model performances.

Model Name	Level	Features Type	Number of Features	CLR	Accuracy	F1 Score	Precision	Recall
Family_ALL_CLR	Family	all	56	yes	0.981 ± 0.004	0.980 ± 0.004	0.987 ± 0.004	0.974 ± 0.006
Family_ALL	Family	all	56	no	0.983 ± 0.004	0.983 ± 0.004	0.989 ± 0.004	0.977 ± 0.006
Family_MW-FDR_CLR	Family	best_fdr	14	yes	0.966 ± 0.004	0.966 ± 0.004	0.976 ± 0.005	0.955 ± 0.006
Family_MW-FDR	Family	best_fdr	14	no	0.970 ± 0.003	0.970 ± 0.003	0.986 ± 0.004	0.955 ± 0.005
Family_MW-Holm_CLR	Family	best_holm	6	yes	0.954 ± 0.003	0.954 ± 0.003	0.957 ± 0.004	0.951 ± 0.004
Family_MW-Holm	Family	best_holm	6	no	0.953 ± 0.004	0.953 ± 0.003	0.951 ± 0.006	0.955 ± 0.004
Genus_ALL_CLR	Genus	all	139	yes	0.990 ± 0.002	0.990 ± 0.002	0.999 ± 0.001	0.981 ± 0.003
Genus_ALL	Genus	all	139	no	0.992 ± 0.002	0.992 ± 0.002	0.999 ± 0.002	0.985 ± 0.003
Genus_MW-FDR_CLR	Genus	best_fdr	32	yes	0.986 ± 0.002	0.985 ± 0.002	0.997 ± 0.003	0.974 ± 0.004
Genus_MW-FDR	Genus	best_fdr	32	no	0.989 ± 0.002	0.989 ± 0.002	0.998 ± 0.003	0.979 ± 0.003
Genus_MW-Holm_CLR	Genus	best_holm	12	yes	0.972 ± 0.003	0.972 ± 0.003	0.982 ± 0.004	0.963 ± 0.004
Genus_MW-Holm	Genus	best_holm	12	no	0.967 ± 0.003	0.967 ± 0.003	0.981 ± 0.004	0.953 ± 0.005

**Table 3 microorganisms-08-01591-t003:** Accuracy of Genus_MW-FDR model on additional projects.

Host Type	Host	Test Project	Accuracy	Number of Samples
human	Homo sapiens	PRJNA385551	1	284
human	Homo sapiens	PRJNA493726	0.932	74
human	Homo sapiens	PRJNA493726_HC	1	19
human	Homo sapiens	PRJNA493726_RA	1	18
human	Homo sapiens	PRJNA493726_SLE	0.865	37
pet	Felis catus + Canis familiaris	PMID32078625	0.983	238
pet	Felis catus	PMID32078625_cat	0.935	46
pet	Canis familiaris	PMID32078625_dog	0.995	192
pet	Canis familiaris	PRJNA401442	1	56
pet	Felis catus	PRJNA470724	0.973	74
pet	Canis familiaris	PRJNA589580	1	35
pet	Canis familiaris	PRJNA592436	1	29

**Table 4 microorganisms-08-01591-t004:** Genus_MW-FDR performance on a mixed class dataset.

Metric Name	Total Dataset	Only Healthy Controls
Accuracy	0.971 ± 0.010	0.988 ± 0.008
Precision	0.976 ± 0.015	0.977 ± 0.015
Recall	0.966 ± 0.016	1.000 ± 0.000
F1 score	0.971 ± 0.010	0.988 ± 0.008

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
