# Peer review of "Pet-Human Gut Microbiome Host Classifier Using Data from Different Studies"

_microorganisms, 2020, doi:10.3390/microorganisms8101591_

Round 1

Reviewer 1 Report

I found this manuscript more interesting for being able to obtain a very accurate classifier using different studies than for the result itself.

I believe that the work can provide the interested scientific community with information on the field of machine learning applied to mcrobiomics data which is still in its infancy. In my opinion, the authors should also have pointed out their microbiological results which can further improve the work done.

Here are some suggestions and changes that I believe should be made before publishing the manuscript.

  1. line 47 you wrote: "It is also generally accepted that species with the same diet are similar at a higher taxonomic level, while host phylogeny reveals itself more at a lower taxonomic level (species and strains)". After this documented statement, choose to use the Gender and Families taxonomic level for the RF model. It is not a contradiction but the reader wonders if it would have been better to toughen the model by choosing the OTU taxonomic level (or better ASV since you use DADA2 for denoising and clustering step) instead of the Genus taxonomic level. Moreover, thinking about the statement in line 47, UNIFRAC is a more logical choice for bDiv (an algorithm that takes into account the phylogeny to calculate the distances) than Bray-Curtis. Anyway, both are not errors, my comment is just a suggestion.

  2. line 89 you wrote: “DADA2 denoising quality parameter value was set to 10”.. Do you refer at the “--p-trunc-q 10 “ parameter? Even if I am a Qiime2 user I didn’t understand that sentence.

  3. About the aDiv calculation, you sampled reads 5 times. It is something that I appreciated a lot. This is something that the majority part of our colleagues do not do but should.
  4. In paragraph line 98 you wrote about “OTU table at genus level”. You seem confused about the idea of OTU. The out_table is a table of reads classified according to the OTU taxonomic level. The Genus table is another taxonomic level. Then the OTU table at the genus level is meaningless. Reading the paragraph, what you have done is just a collapse of OTUs to create a counts table at the different taxonomic levels.  I invite the authors to correct this part to avoid confusion in the readers. Also, it should be better to use the term ASVs instead of OTUs. The OTUs imply the use of algorithms of clustering (i.e. UCLUST) where the percent of identity is required (usually 97% or 99%).  
  5. In the sub-chapter “Filtering of low abundance OTUs”, you use the term “abundance” in the wrong way. If I well understood, you meant occurrence and not abundance. They have very different meanings. The taxa occurrences are something very tricky to manage. The high presence of zeros (peculiar characteristic of this type of NGS data namely “sparsity of data”) has two different meanings. Zeros called "rounded zeros” are a direct consequence of sub-sampling and therefore highly dependent on the performance of the sequencing platform. On the other side, zeros called "essential zeros" which truly represent the absolute absence of that composition in the observation dependent on the biological sample. This latter typology of zero not dependent on the library size, therefore, maintains intrinsic biological information.
    To treat the round zero, being a problem due to technical “detection limits”, would be reasonably replacing it by a suitable value smaller than the minimum value possible (represented by 1 read) like 1/2. Although, this method has been developed as a traditional approach called imputation procedure, using arbitrary replace strategy requests prudence. It must not distort the general structure of the data, in particular, the covariance and metric structure of the features (Martín-Fernández J, Barceló-Vidal C, Pawlowsky-Glahn V. Dealing with zeros and missing values in compositional data sets using nonparametric imputation. Math Geol 2003;35:253e78.).
    Then I invite you to replace the term abundance with occurrence.
  6. Re-formatting table 2.
  7. Line 191- Bacteroidetes is a Phylum.. so you should write: families belonging to Bacteroidetes phylum.
  8. You should expand the discussion session. You should try to explain:
    - The non-improvement showed by the usage of CLR transformation is great information for the microbiome field. You should stress/underline this information.
    -you found differential taxa between different organisms, you should try to explain why in you think Verrucomicrobia is present just in human. What advantages does the human gut offer to this class of bacteria?
    • Regarding the model, you should explain your point of view that Holm and FDR correction show such big differences.
  9. In Table 3 you show several models with accuracy 1… I am very surprised about the possibility to have a classification model with 100% True prediction looking at 284 samples. If I were you I would double-check these results very carefully... If there are no mistakes, you should emphasize this information in the discussion.

Author Response

  1. Here we decided to use a genus taxonomy level for making the classifier in order to compare the obtained results with previous studies. Specifically, differences at a genus and a family level were previously described between human and animal microbiomes. Moreover, feature dimension reduction prevents overtraining of the model. Bray-Curtis dissimilarity index is commonly used in a microbiome research (10.1128/mSystems.00031-18 ; 10.1016/j.schres.2018.09.014; 10.1038/s41598-017-13601-y). Moreover, BC measure is highly correlated with weighted UNIFRAC (10.1038/ismej.2012.88). Since we used our own custom reference database, constructed from SILVA and restricted to gut microbiome microorganisms, BC was chosen as a suitable measure for beta-diversity calculation.
  2. Yes, we refer to the --p-trunc-q 10 parameter. We now made it explicit in the text.
  3. Thank you. Indeed, alpha diversity is very sensitive to rare species, and sampling helps to deal with it.
  4. Thank you for noticing, the usage of the OTU term was indeed misleading. Since QIIME2 was used, there are not OTUs, but ASVs, summed by different taxonomy levels. We have now changed it. We have now removed the OTU term from the text.
  5. We agree with the reviewer. We have changed the term ‘abundance’ to ‘occurrence’. However, in our study we do not distinguish between true zeros and low abundant species as both things would result in low occurrence in the full dataset. We merely wanted to restrict model features to the most meaningful ones, and the features mostly filled with zeros are not useful for prediction.
  6. Thank you for noticing, we have reformatted the Table 2.
  7. Thank you for noticing, we have changed this in the text.
  8. Thank you, we added the discussion of these issues in the text. I also explain it here:
    1. The fact that the usage of CLR transformation did not significantly improve results in our case might be because the prediction accuracy is good with both methods.
    2. The main, if not to say the only, Verrumicrobia member in a gut microbiome is Akkermansia genus. Akkermansia bacteria are known to live in a mucin layer and degrade it. Previous studies show that indeed, feline and canine gut microbiome lack Akkermansia because this ecological niche is occupied with other species, specifically members of Bacteroidaceae, Prevotellaceae, Clostridiales, Faecalibacterium, and Fusobacteria phylum (10.3390/vetsci7020044).
    3. The Holm-Bonferroni procedure is more strict, while FDR correction yields more features. A RF model using more meaningful features allows for better prediction.
  9. We have checked the data as you suggested. It was not a mistake, and all the 284 samples were predicted as human, with probabilities of state 1 (human) ranging from 0.67 to 1.0. We associate it with the fact that all the samples in this project were healthy controls. Other human projects included individuals with SLE (systemic lupus erythematosus) and RA (rheumatoid arthritis), and the total accuracy was lower due to that. But the algorithm is not perfect, the perfect recall on the healthy control samples is balanced with precision equaling 0.98, it is specified in Table 4.

Reviewer 2 Report

Major Comments

Bykova et al. present an interesting solution to an unusual problem - that of people submitting microbiome samples from non-human hosts to facilities that are intended to process samples from human hosts. The most scientifically significant contribution of their work is the nice cross-study cross validation that is performed. They find patterns that hold irrespective of the study in which they were found which in my opinion greatly lifts the significance of the work.

My main criticism of the work is that there are several tasks where the authors have developed tools that are freely available elsewhere. Mostly they have done a good job but in one case this results in changes being required to manuscript.

The authors analyse differential abundance between samples using Mann-Whitney tests applied to median OTU counts. Applying two-sided Mann-Whitney tests in this way ignores the compositional count nature of the data and can lead to, for example, high false positive rates (see, eg. https://doi.org/10.1371/journal.pcbi.1003531). There are several good tools for differential abundance testing that should be used to replace these tests. Some examples are ANCOM (now  available in several variants), ALDEx, and DESeq2 (see the dada2 tutorials).

The title and abstract are a little bit misleading. The general term "pet" implies more than cats and dogs, but only samples from cats and dogs were tested in the study. I don't think that this was intentionally misleading, but it would be better to be more explicit.

The final major change that is required is that the methods section is, in general, very poorly referenced. The following things need to be referenced:

  • Chao diversity
  • dada2
  • QIIME2
  • QIIME2 feature-classifier
  • Holm and FDR (these are standard but it can't hurt - some people use different terms)
  • skbio
  • Random Forest learning

Minor Comments

Of lesser importance is that the authors could have saved themselves some time by downloading > 500 dog and cat samples and many thousands of human samples from Qiita using redbiom, and performed sample classification using q2-sample-classifier, which provides RF classifiers for the purpose. The methods they have used to acquire and process data from NCBI are fine, however.

Some minor corrections

P1L8 Change "is usually" to "can be"

P1L9 delete "the" from "the environmental"

P1L14 change "microbiota" to "microbiomes"

P1L24 Change "The microbiome" to "A microbiome".

P1L25 Change "Microbiome" to "Microbiota"

P1L35 Change "The other" to "An other"

P2L47 Change "taxa" to "taxonomic"

P2L53 Delete "taxa" and change "microbiome" to "microbiome taxonomic composition"

P5L143 Change "github" to "GitHub"

Author Response

  1. Mann-Whitney (MW) test is a non-parametric test widely used in the microbiome research (10.1186/s40168-017-0237-y), mostly for the robust preliminary feature filtering. The dataset used in our study is highly heterogeneous since it originates from different study projects. So applying ANCOM, ALDEX or Deseq2 directly on the mixed data set could bring bias even if the projects were balanced. Therefore we chose to apply MW test to the median abundance values of each project to exclude any possible artifacts. Though we understand that MW is less sensitive and has smaller power, than the tests specifically designed for compositional data.
  2. Thank you for the suggestion. We wanted to make the title short and assuming that cats and dogs are the most common pets we chose the present title. We can suggest another title that would be more explicit, for example “Human vs Canine and Feline gut microbiome host classifier using data from different studies”. Does this version make more sense?
  3. Thank you for pointing this out. We added the missing references in the text.
  4. We also fixed all minor text corrections suggested by reviewer 2.